

# Detecting breakpoints in global temperature

Junbo Duan[†], Lanling Zhao[†], Qing Wang[‡], and Pei Li[§]

[†]Key Laboratory of Biomedical Information Engineering of Ministry of Education and the Department of Biomedical Engineering, School of Life Science and Technology, Xi'an Jiaotong University, Xi'an, 710049, China
[‡]School of Electronic Engineering, Xidian University, Xi'an, 710071, China
[§]School of Earth Science, Zhejiang University, Hangzhou, 310027, China

**Correspondence:** Junbo Duan (junbo.duan@mail.xjtu.edu.cn)

**Abstract.** This paper aims to address the continuous debate of whether a 'hiatus' occurred in the turn of this century. Several models have been employed to fit the global mean surface temperature data, and the results suggest that the allegation of an occurrence of 'hiatus' lacks statistical evidence. However, these models had potential deficiencies in their capacity for detecting breakpoints, thereby weakening the arguments that deny the existence of a 'hiatus'. To address this issue, we propose an

improved sparse representation model, which can automatically segment and fit temperature records using piecewise polynomials. Simulations revealed improved detection performance; studies on five prominent global temperature records produced 2 to 6 breakpoints, none of which occurred after the year 1976, thus reinforcing arguments denying the existence of a 20th century 'hiatus'.

## 1 Introduction

Because of its broad potential to impact human activities, global warming receives widespread attention from both the scientific and public sectors of society. In the last few decades, there has been a continuous debate on whether a 'hiatus' occurred in

the global mean surface temperature (GMST) (Carter, 2006; Medhaug et al., 2017; Yan et al., 2016). Since the term 'hiatus' is strongly misleading, the synonyms 'pause', 'slowdown', and 'stop' are also used in the literature (Lewandowsky et al., 2016). The 'global warming hiatus' was defined as a "reduction in GMST trend during 1998-2012 as compared to the trend during 1951-2012" (Working Group I to the Fifth Assessment Report of the Intergovernmental Panel on Climate Change, 2013), implying that there was an abrupt change in the GMST trend during the last years of the previous century.

Due to the complicated nature of earth system dynamics (internal weather factors such as ocean circulations and atmospheric motions, *e.g.,* El Niño/southern oscillation (ENSO), volcanic eruptions and aerosol; external factors such as solar irradiance; anthropogenic factors such as greenhouse gas emissions), long-term changes (*i.e.,* trends) are difficult to distinguish from short-





term changes (*i.e.,* fluctuations) (Foster and Rahmstorf, 2011). While nearly one hundred papers have been published on this debate (TI=((climate change OR global warming OR global temperature) AND (hiatus OR slowdown OR pause)) in the Web
of Science database), very little statistical evidence supports the existence of a 20th century GMST 'hiatus'.

Most of these studies (Foster and Abraham, 2015; Cahill et al., 2015; Rahmstorf et al., 2017) primarily employed methods that had insufficient detection abilities given that the detection of a 'hiatus' is sensitive to the way a time series is processed (Santer et al., 2011; Hawkins et al., 2014), *e.g.,* fix-sized windows (Fyfe et al., 2016), and were prone to type I errors when there were numerous changes. Therefore, highly sensitive and automatic segmenting methods are required to detect a
30 'hiatus' in the data. Cahill et al. (2015) and Rahmstorf et al. (2017) adopted the change points (CP) model from the statistical society to detect trend changes. A CP is formally defined as the point in a dataset where the first order difference changes, and is detected by fitting the GMST data with piecewise linear lines. However, since the CP model imposes a continuity constraint, its detection sensitivity is degenerated. In addition, the CP model requires the number of CPs being fixed *a priori*, other methods (Fyfe et al., 2016) arbitrarily fix the size of windows. Third, the CP model assumes a linear trend, which further limits its
detection power. Last, the solving of a CP model is based on a Markov chain Monte-Carlo (MCMC), which suffers from the fact that its convergence to a global solution is not guaranteed.

In this paper we present a new sparse representation model with four improvements: (1) By abandoning the continuity constraint, breakpoints are used instead of CPs, such that extra degrees of freedom are available. This modification is not only reasonable from a theoretical point of view but is also supported by the temperature records and thus improves the
40 model's detection power. (2) By utilizing a prominent model selection method, *i.e.,* Bayesian/Schwarz information criterion (BIC/SIC) (Schwarz, 1978), the number of breakpoints is determined automatically. (3) The model fitting is extended from first order (or linear) to higher orders, such that other trend changes can be detected. (4) Dynamic programming is employed to solve the proposed model such that a global solution is guaranteed.

## 2   Methods

First, for a signal (or time series in our case) $\boldsymbol{y} = [y_1, y_2, \ldots, y_N]^T$ of length $N$, we define the term 'breakpoint' to mean the location where two consecutive pieces (or segments) breaks. *e.g.,* the $k$th breakpoint $v_k$ divides the $k$th piece $(v_{k-1}, \ldots, v_k]$ and the $k+1$th piece $(v_k, \ldots, v_{k+1}]$. We also denote $\boldsymbol{v} = [v_1, v_2, \ldots, v_K]$ as a set of $K$ breakpoints. For convenience, $v_0 = 0$ and $v_{K+1} = N$. Here, the two pieces are distinguished by their statistical distributions (*e.g.,* the amplitude for the piecewise constant signals, the baseline for the piecewise linear signals, *etc.*). We denote

$$50 \quad \varepsilon_k = \sum_{i=v_{k-1}+1}^{v_k} (y_i - x_i)^2 \qquad (1)$$

as the fitting error of the $k$th segment and $x_i, i \in (v_{k-1}, v_k]$ is the least squares fitting of the $k$th segment. $x_i$ can be a polynomial of any order $r$: constant ($r = 0$), linear ($r = 1$), quadratic ($r = 2$), *etc.*





It is worth mentioning that the data fit is highly dependent on segmenting. If the configuration of segmenting is known, *i.e.,* the breakpoints $\boldsymbol{v}$ is fixed, all $\varepsilon_k$'s can be estimated up to $K$ least-square fittings. As a result, the fitting of $\boldsymbol{y}$ with piecewise

polynomials of order $r$ can be formulated as the following breakpoint detection problem

$$\arg\min_{\boldsymbol{v}} \left\{ \sum_{k=1}^{K+1} \varepsilon_k \right\}. \tag{2}$$

However, this problem is not well defined due to the length of $\boldsymbol{v}$ (*i.e.,* $\|\boldsymbol{v}\|_0$ or $K$, here $\|\cdot\|_0$ is the $\ell_0$ quasi norm) not being considered. Since any consecutive $r + 1$ data points can be fitted with a polynomial of order $r$ with zero fitting errors, there is at least one $\boldsymbol{v}$ with length $\|\boldsymbol{v}\|_0 = K_{\max} = \lceil \frac{N}{r+1} - 1 \rceil$ that yields a zero value objective function. When the length is large

enough, there are a great many of $\boldsymbol{v}$ that yield zero values. Therefore, we must take the length of $\boldsymbol{v}$ into account by penalizing each breakpoint with a cost $\lambda$; consequently and thus the penalized least squares optimization problem reads

$$\arg\min_{\boldsymbol{v}} \left\{ \sum_{k=1}^{K+1} \varepsilon_k + \lambda \|\boldsymbol{v}\|_0 \right\}. \tag{3}$$

This problem is an incarnation of the Occam's razor principle, or the law of parsimony, and is familiar to signal processing community as a sparse model (Eldar and Kutyniok, 2012).

The penalty parameter $\lambda$ can control the fitting's quality and sparsity, and hence should be tuned carefully. Within the Bayesian framework, $\varepsilon_k$ can be viewed as the maximal likelihood estimation when Gaussian noise is present (Idier, 2008). Therefore, various theories of model selection can be used to choose a proper $\lambda$ (Stoica and Selén, 2004); the Akaike information criterion (AIC) (Akaike, 1974), the Bayesian/Schwarz information criterion (BIC/SIC) (Schwarz, 1978), the Hannan and Quinn criterion (HQC) (Hannan and Quinn, 1979), the minimum description length (MDL) (Rissanen, 1983), and other

variants (Markon and Krueger, 2004) can be directly employed.

When $\lambda$ is fixed, an unsophisticated method to solve (3) is through a brute-force search that tests all the combinations of breakpoint locations exhaustively, *i.e.,* $C_{N-1}^K, \forall K \in [0, K_{\max}]$ of possible $\boldsymbol{v}$'s, which is computationally prohibited when $N$ is larger than 50. Advanced methods were developed to reduce the computational burden. For more detailed explanation of the rational of this model and the associated optimization algorithm, the readers are referred to (Duan et al., 2019) and the

references therein.

## 3  Results

### 3.1  Justification of the breakpoints

The first enhancement to the presented model is through its utilization of breakpoints. Breakpoints are discontinuities in a fitting, and their use has been controversial among previous studies. The CP model assume that the fitting is continuous in

order to be 'physical'. However, as shown in Fig. 1, the annual variance, which is defined here as the difference between the





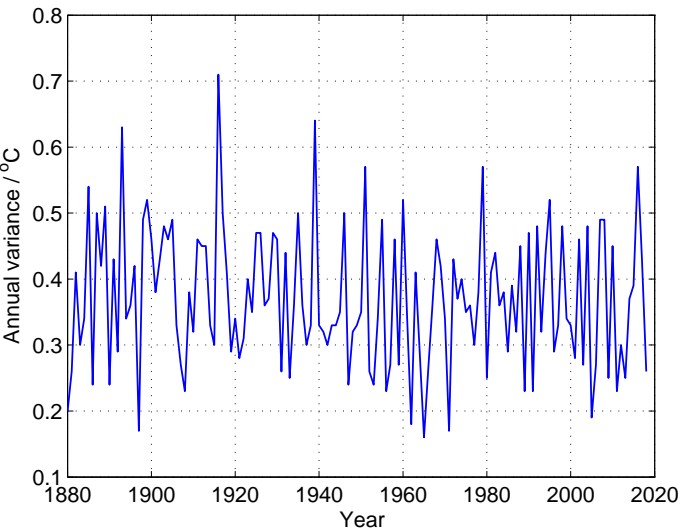

**Figure 1.** Annual variance in GISTEMP dataset.

maximal and minimal monthly temperature within a year, is so immense ($0.37 \pm 0.10°$C) that it cannot be ignored. For a larger time space, the assumption that *annual temperatures are discontinuous* makes sense.

### 3.2 Performance of the proposed model

To test the performance of the proposed model, and compare it with the CP model in (Cahill et al., 2015), we simulated a dataset and fed the data to the proposed model and the CP model (code available at http://iopscience.iop.org/1748-9326/10/8/084002/media/Rcode_CPA.R) as well. Details of the simulations are presented in Sec. I of the supplementary materials. The simulations suggest that the proposed model can achieve higher sensitivity and specificity compared to the CP model when the breakpoints and the slope are significant.

### 3.3 Fitting of Global surface temperature

Five data sources were used in our analysis: the GISS surface temperature analysis (GISTEMP) from the National Aeronautics and Space Administration (NASA) (Team, 2018), the temperature records of the National Oceanic and Atmospheric Administration (NOAA) (Oceanic and , NOAA), HadCRUT4 from Met Office Hadley Centre Climatic Research Unit (Morice et al., 2012), the Cowtan and Way's study (Cowtan and Way, 2014), and Berkeley earth (Rohde et al., 2013). The former two datasets record the global land-ocean temperatures since the year 1880, and the latter three datasets since the year 1850.

We fitted the five time series using our proposed method. Both polynomial order $r = 1$ and 2 were used for least squares fitting. Because of its robust performance (Duan et al., 2019), BIC/SIC was used to determine the penalty parameter $\lambda =$



$(r+1)\sigma^2 \ln N$, where $N$ is the length of the time series, and $\sigma$ ( estimated from the last 50 points of each dataset) the standard deviation of the noise.

Fig. 2 shows the breakpoint detection results of two representative datasets, namely, the GISTEMP and HadCRUT4; one includes coverage of the whole globe, and the other has a large gap of missing data in the Arctic (Rahmstorf et al., 2017). Because of space limitations, the full results of the five datasets are presented in Sec. II of the supplementary materials. Tab. 1 summarizes the breakpoints and amplitudes, the standard deviation of the residual, and the $p$-value of the one-sample Kolmogorov-Smirnov test of the residuals of all five datasets with the different polynomial orders $r = 1, 2$. Fig. 3 shows the distribution of all the detected breakpoints, with absolute amplitudes $0.20 \pm 0.07°C$, a maximum temperature of $0.34°C$, and a minimum temperature of $0.07°C$.

From these results, we can draw the following conclusions arranged in chronological order: (1) two dramatic climate coolings were detected around the year 1901 and 1945 in all the datasets except HadCRUT4; (2) a continuous warming between the year 1901 and 1945 is observed in all the datasets except HadCRUT4; (3) a slow cooling is observed in the years from 1936 to 1976 from HadCRUT4, CW, and Berkeley (see panels (c), (g), and (i) of Fig.2 in the supplementary materials), or a mixture of breaks are observed from GISTEMP and NOAA; this is called the 'big hiatus' in other studies (Cahill et al., 2015; Fyfe et al., 2016; Carter, 2006); (4) there is no breakpoint after the year 1976, suggesting that no detectable 'hiatus' occurred in the turn of the century as claimed in some literature; (5) overall, the second order fittings of GISTEMP and NOAA provide a concise and meaningful trend for the GMST (see panels (b) and (f) of Fig.2 in the supplementary materials).

## 4   Conclusions

In this paper, a piecewise polynomial model is proposed to detect the breakpoints in the GMST records, and the results show that no breakpoints are found after the 1970's. Thus, there is no support for the existence of a 'hiatus' in global warming after the turn of the century, which is consistent with other studies; there findings are useful as supplementary evidence for the 'hiatus' debate.

The highlights of the proposed method are twofold. First, since the selection of a time interval is important for curve fitting and trend estimation (Santer et al., 2011), the proposed method combines segmentation of the time intervals and data fitting in a natural and automatic way, thereby reducing the bias introduced by manual selection of a time interval. Second, the proposed method has higher detection sensitivity compared with the other methods, and negative results further reduce the chance of missing the true 'hiatus'.

As independent researchers, we conclude that the supposed occurrence of a recent 'hiatus' lacks statistical evidence, or was of less significant than other historical 'hiatuses' since the 1850s.





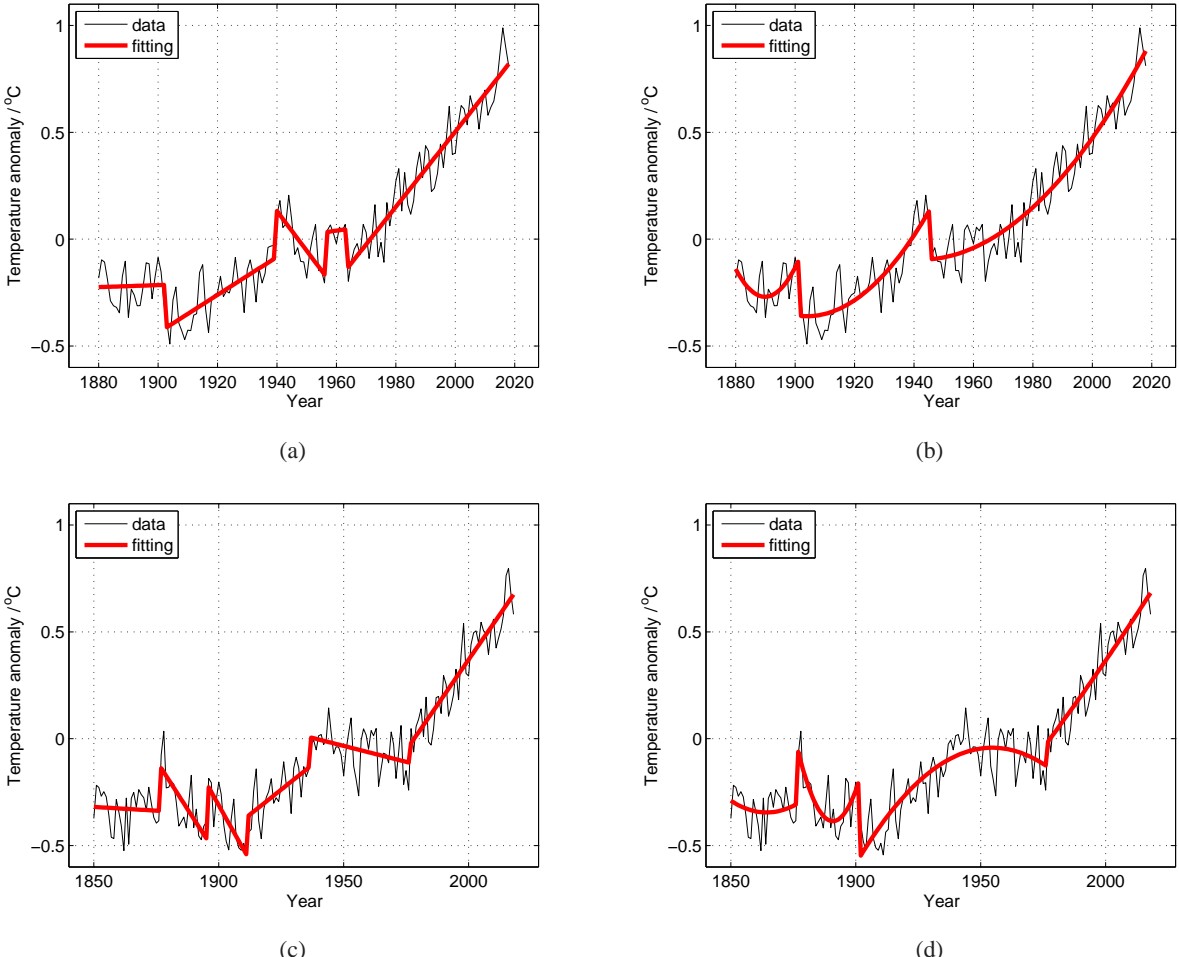

**Figure 2.** The breakpoints of in the global temperature. Left and right panels show results with polynomial order $r = 1$ and $r = 2$, and the top and bottom panels the GISTEMP and HadCRUT4 datasets, respectively. Full results of the five databases are shown in Figure 1 of supplementary materials.

*Code and data availability.* Data are available from the websites cited therein, and codes are available upon request.

*Author contributions.* JD, QW, and PL designed this study. JD and LZ wrote the code for the study. JD wrote the manuscript, QW and PL revised the manuscript. All have read the manuscript and approved the final version.

*Competing interests.* The authors declare that they have no competing interests.

*Acknowledgements.* This work was supported in part by the National Science Foundation of China under Grant 61771381 and Grant 61401352.



**Table 1.** Summary of the fitting results of five datasets.

| Dataset | Order ($r$) | Breakpoint ($v_k$)/amplitude ($x_{v_k+1} - x_{v_k}$) | Residual ($\sigma$) | $p$-value |
|---|---|---|---|---|
| GISTEMP | 1 | 1902/-0.198,1939/0.225,1956/0.199,1963/-0.176 | 0.086 | 0.736 |
| | 2 | 1901/-0.254,1945/-0.223 | 0.089 | 0.877 |
| NOAA | 1 | 1895/0.206,1911/0.189,1932/-0.082,1945/-0.344,1963/-0.167 | 0.075 | 0.937 |
| | 2 | 1901/-0.236,1945/-0.247 | 0.084 | 0.748 |
| HadCRUT4 | 1 | 1876/0.199,1895/0.238,1911/0.181,1936/0.141,1976/0.091 | 0.086 | 0.881 |
| | 2 | 1876/0.247,1901/-0.337,1976/0.112 | 0.090 | 0.682 |
| CW | 1 | 1876/0.305,1883/-0.127,1901/-0.219,1936/0.156,1976/0.077 | 0.086 | 0.940 |
| | 2 | 1876/0.215,1901/-0.258,1944/-0.123 | 0.088 | 0.756 |
| Berkeley | 1 | 1864/0.229,1876/0.318,1887/0.181,1902/-0.199,1936/0.159,1976/0.074 | 0.081 | 0.887 |
| | 2 | 1862/0.192,1876/0.343,1901/-0.273,1944/-0.101 | 0.083 | 0.845 |

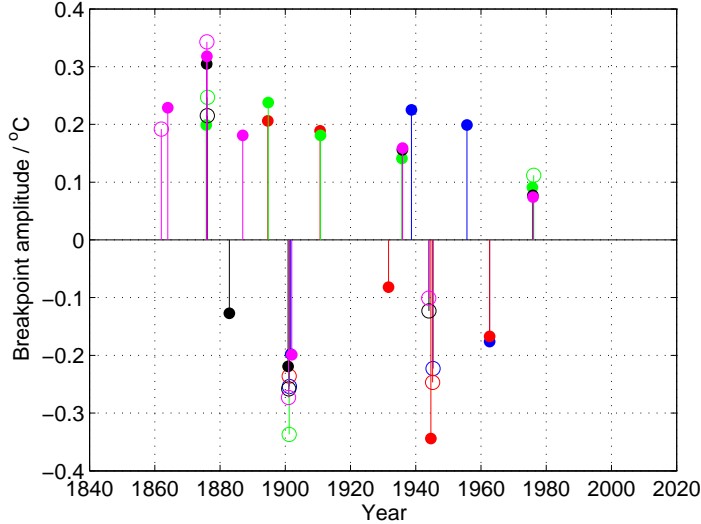

**Figure 3.** The distribution of breakpoints. Dot and circle markers represent polynomial order $r = 1$ and 2, respectively. Blue, red, green, black and magenta colors represent dataset GISTEMP, NOAA, HadCRUT4, CW, and Berkeley, respectively.





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
