# Peer review of "Detecting breakpoints in global temperature"

_Earth System Dynamics, 2019_

## Referee Comment (RC1) · Anonymous Referee #1 · 15 Sep 2019

The authors used an improved sparse representation model to fit global mean surface temperature time series to test the existence of global warming hiatus. They found 2 to 6 breakpoints which all occur before1976. Hence they denied the existence of recent global warming hiatus during 1998-2012.

Though I appreciate the authors' improved sparse representation model, I am not convinced by their results. Thereby, I cannot recommend publication of this manuscript at Earth System Dynamics.

First, the examination of global warming hiatus should not be limited to the analysis of one-dimensional time series of global mean surface temperature. Surface temperature observations (plus subsurface ocean and atmosphere), reanalysis data and climate model simulations provide rich three-dimensional information that can used to infer the physical mechanisms of global warming hiatus. Several mechanisms (such as negative

phase of the Interdecadal Pacific Oscillation and enhanced heat uptake in the Atlantic) have been proposed to explain the recent hiatus. As such, it is very hard to deny the existence of recent hiatus simply based on some fitting of global mean surface temperature time series.

Besides, surface temperature observations have great uncertainties. An uncertainty maximum is obvious around World War II. Many factors, such as quality control, bias correction and differences between ship and buoy data, all can contribute to the uncertainties of the surface temperature datasets used in this study. Thereby, the authors need to consider these observational uncertainties in their analysis.

---

## Referee Comment (RC2) · Anonymous Referee #2 · 29 Oct 2019

In this manuscript, the authors used statistical models to detect the breakpoints in multiple global mean surface temperature products. The authors' argued that their methods showed advantages over published results. The authors' results do not support a global warming hiatus during the 20th century.

I find that this manuscript, in its current form, does not fit the scope of Earth System Dynamics and lacks appeal and relevance to the climate science community in general. Please see my comments bellow.

Major comments: 1. This manuscript lacks physical understanding of statistical method and results. The statistical models deny a 20th century "hiatus". How about the other breakpoints your method does pick up? Do they have physical meaning?

2. This study lacks detailed comparison with previous results, in order to illustrate the advantage of the authors' new method. In the supplemental information, the authors

compared their method and the CP method using a synthetic data. But, a comparison with previous methods using real data is more meaningful. Time series of the global mean surface temperature have rich physical meaning, reflecting impacts from internal interannual/decadal variations and forced responses to volcano eruptions and anthropogenic greenhouse gas and aerosol emissions. A pure statistical fitting without physical examination and explanation is insufficient to understand the rich information in historical temperature evolution.

Minor comments: 1. Line 2 in Abstract: please specify that you are using statistical models, instead of comprehensive climate models. Please also pay attention to the use of "model" throughout the manuscript, as readers of Earth System Dynamics may mistake models as "climate models".

2. Line 20–23: please re-write. "ENSO" is not an internal weather factor. Also, it is best to categorize volcanic eruptions as external climate forcing. Aerosols have anthropogenic component.

3. Line 23–15: please list the date when the search was done.

4. Line 33–34: please re-write. Add "and" between two clauses?